# Urinary Incontinence in Physically Active Older Women of Northeast Brazil

**DOI:** 10.3390/ijerph18115878

**Published:** 2021-05-30

**Authors:** Thais Sousa Rodrigues Guedes, Marcello Barbosa Otoni Gonçalves Guedes, Hévila Kilvia Miguel de Oliveira, Rodrigo Lopes Soares, Vitor Leandro da Cunha, Johnnatas Mikael Lopes, Nayara Priscila Dantas de Oliveira, Javier Jerez-Roig, Dyego Leandro Bezerra de Souza

**Affiliations:** 1Graduate Program in Health Sciences, Center of Health Sciences, Campus Universitário Lagoa Nova, Federal University of Rio Grande do Norte (UFRN), Natal 59078-970, Brazil; thais.sousarodrigues@gmail.com; 2Department of Physiotherapy, Campus Universitário Lagoa Nova, Federal University of Rio Grande do Norte (UFRN), Natal 59078-970, Brazil; marcelloguedes21@hotmail.com; 3Faculty Mauricio de Nassau, Av. Engenheiro Roberto Freire, Natal 59078-600, Brazil; kilvia.oliveira22@outlook.com (H.K.M.d.O.); contato-rodrigo@outlook.com (R.L.S.); 4Graduate Program in Neuroengineering, Edmond and Lily Safra International Institute of Neuroscience, Santos Dumont Institute, Macaiba 59280-000, Brazil; vitor.cunha@edu.isd.org.br; 5Department of Medicine, Federal University of Vale do São Francisco (UNIVASF), Paulo Afonso 48605-560, Brazil; johnnataslopes2@gmail.com; 6Graduate Program in Collective Health, Department of Public Health, Federal University of Rio Grande do Norte (UFRN), Natal 59078-970, Brazil; nayoliveira.fisio@gmail.com; 7Research Group on Methodology, Methods, Models and Outcomes of Health and Social Sciences (M_3_O), Faculty of Health Science and Welfare, Centre for Health and Social Care Research (CESS), University of Vic-Central University of Catalonia (UVic-UCC), 08500 Vic, Barcelona, Spain; dysouz@yahoo.com.br; 8Graduate Program in Health Science, Department of Public Health, Campus Universitário Lagoa Nova, Federal University of Rio Grande do Norte (UFRN), Natal 59078-970, Brazil

**Keywords:** elderly, loss of bladder control, exercise

## Abstract

Low- and moderate-impact physical activity (PA) is associated with the prevention of urinary incontinence (UI). The objective of the cross-sectional study presented herein is to analyze the factors associated with UI in physically older active women who participate in senior community groups. The variable UI was measured by the International Consultation Incontinence Questionnaire Short Form (ICIQ-SF). Socioeconomic variables were also collected, along with data on life habits and clinical history. The multivariate analysis employed Poisson’s Regression with robust variance for factors associated with UI. Of the 106 participants evaluated, 54.7% presented UI, of which stress incontinence was more frequent, with 40.6%. UI presented a statistically significant association with dizziness/loss of balance during Activities of Daily Living (ADL) (prevalence ratio-PR 1.48; 95% CI 1.06–2.07) and nocturia (PR 1.63; 95% CI 1.05–2.55). Despite PA being a protection factor, UI presented an elevated prevalence in the older population, and therefore, other biological, social, and cultural aspects could also contribute to the occurrence of UI in this age group. Moreover, physically active older women with UI presented nocturia and dizziness/loss of balance during ADL, regardless of education levels and the number of births. These findings can help improve multi-professional programs aimed at promoting, preventing, and managing UI in the public.

## 1. Introduction

The global older population has been increasing significantly by 3% per year. By 2050, the population will be constituted by approximately 2.1 billion seniors [1]. In Brazil, currently, the proportion of people over 60 years old is 13%, and by 2050 [2], it is estimated that seniors will constitute 29.3% of the population [3]. In this context, optimization of the public services offered to this share of the population is essential, with integral actions and public health policies supported by adequate epidemiological information [4].

Considering the physiological changes that accompany the advance in age, it is important to consider strategies oriented to maintaining quality of life (QL) and promoting older people’s health. This can lead to a more independent older population, contributing to better performance and self-confidence when carrying out Activities of Daily Life (ADL). It must be highlighted that healthy aging refers to developing and maintaining a functional capacity so that older people can embrace wellness as a part of everyday life [5]. The term *active aging* is associated with healthy aging and is used by the World Health Organization (WHO) to refer to the process of optimizing health, participation, and safety opportunities to improve the quality of life [6]. Studies have demonstrated the importance of physical activity (exercise) for good physical conditioning of seniors, leading to improvements in abilities, muscular strength, mobility, flexibility, and cognition, promoting functional improvements that favor good QL in older people [7].

Considering the relevance of practicing physical activities to achieve good structural conditioning during the aging process, it must be noted that the regular practice of exercise is fundamental at all stages of life [8]. Keeping active while aging can be beneficial in promoting good health and preventing some issues, such as urinary incontinence (UI), especially in older women [5].

The International Continence Society defines UI as the complaint of involuntary loss of urine, which affects approximately 25–45% of the female population throughout life [9]. UI is pointed to as one of the greatest concerns of health professionals, being one of the dysfunctions that most affect women nowadays, causing negative impacts to QL [10].

Studies have pointed out that low- to medium-impact physical exercise can help prevent dysfunctions in the pelvic floor, especially UI [11,12]. It is suggested that physically active women have stronger pelvic floors; also, with the increase in abdominal pressure during physical exercises, there is a reflex contraction of pelvic floor muscles, which contributes to the urinary continence mechanism [11,12]. Therefore, older women can benefit from this strategy.

High-impact physical exercises are described in scientific literature as a risk factor for UI. High impact activities can affect the continence mechanism due to a change in the force transmitted to the pelvic floor, increasing intra-abdominal pressure significantly. These changes compromise the supporting, suspension, and contention mechanisms of the pelvic floor, which suffers intense and repetitive overload, leading to its weakening [13].

UI is a problem that directly affects the QL of older women, either due to functional or psychological limitations that lead to restrictions in social life [14]. The majority of studies on factors associated with UI in physically active older people have been carried out in developed countries, with different socioeconomic and health-related characteristics from the Brazilian Northeast. Therefore, the objective of this study is to identify the factors associated with UI in physically active older women living in Northeast Brazil.

## 2. Material and Methods

### Participants and Procedures

This is a quantitative, cross-sectional study with a descriptive approach and inferential analysis. Data collection occurred between September and December 2017. A non-probabilistic sample of older women who practiced regular physical exercise in two community groups of Natal, Northeast Brazil, was obtained. The senior groups were selected for convenience; both offered regular low-impact supervised physical activities, five times a week: hydrogymnastics, dance, and gymnastic routines, for 60 min each.

The sample size of this study (*n* = 106) enabled the identification of a 17% difference of proportion, with a confidence level of 95% and 80% test power [15]. The inclusion criteria were: women at least 60 years old [6], who practiced physical exercise at least two times a week (any type offered by the community groups). The exclusion criteria were: women with a cognitive deficit preventing the understanding of the questionnaires applied during the interviews; women who could not attend the interview; women who, for any reason, did not complete the questionnaires; and those who did not want to participate in the study.

This study was approved by the Research Ethics Committee of the Maurício de Nassau University Center, protocol number 2.284.118. All participants had voluntarily agreed to participate and signed Free Informed Consent Forms.

Data collection occurred during individual interviews with the participants, which lasted about 30 min, in a calm, reserved, and appropriate location. All questions were explained to the interviewees by two trained interviewers. Initially, a semi-structured questionnaire was employed, with open-ended and closed-ended questions, to obtain personal information from the participants: name, telephone number and address, socioeconomic history including age (stratified according to the age groups defined by the Brazilian Institute of Geography and Statistics, (IBGE)), Body Mass Index (BMI), race, marital status, education level, family income as stratified by the Brazilian government in terms of the minimum wage in 2017, access to health services, life habits (consumption of tobacco or alcohol, and the frequency of physical activity), obstetric history (number of pregnancies and births), presence of comorbidities or multimorbidities (diabetes, systemic arterial pressure, arthritis/arthrosis), use of medications, type of UI, dizziness and loss of balance during ADL (have you felt dizzy or lost your balance in daily activities?), falls due to dizziness or loss of balance during ADL (have you fallen due to dizziness or loss of balance during daily activities?), proportion of restful sleep (little, moderate, complete), and occurrence of sleep disruption from having to urinate during the night (nocturia). The criteria to evaluate the practice of physical activity were based on the questions within Module P-Lifestyles, from the questionnaire of National Health Research, which addresses research involving participants from the Brazilian population [16]. According to this questionnaire, physically active women were defined as those who practiced any type of physical exercise or sport in the last three months (not considering physiotherapy). All questions were answered based on self-reporting by the interviewee. The aforementioned variables are organized in Tables 1 and 2.

Measurement of UI employed the International Consultation Incontinence Questionnaire Short Form (ICIQ-SF). This questionnaire consists of four questions that assess the frequency and amount of urine loss (evaluated by two different questions). The ICIQ-SF score is obtained by adding the answers to questions 3, 4, and 5 of the questionnaire, and varies between 0 and 21 points. The higher the score, the higher the severity and impact of UI on QL [17]. Based on the score, the questionnaire quantifies the impact of UI on QL, and is divided into no incontinence (0 points), slight impact (1–5 points), moderate impact (6–12 points), severe impact (13–18 points), and very severe impact (19–21 points) [17]. Finally, the situations during which urine losses occur were investigated, such as during the practice of physical exercise, or when coughing or sneezing, and encompassed a set of eight items for self-diagnosis, related to the causes or UI situations experienced by participants [17,18].

The information collected with the questionnaires was organized in a databank. Statistical analysis employed software Stata 13.0 (StataCorp LLC, College Station, TX, USA). For the quantitative variables, data adequacy to the normal distribution was assessed by considering established criteria: minimum and maximum values contained within the interval corresponding to the mean value ± three standard deviations (SD), and modules of kurtosis and asymmetry lower than twice their corresponding SD. Initially, results were presented by descriptive statistics, with tables for absolute and relative data. Then bivariate analysis was carried out, using Pearson’s chi-squared test or Fisher’s Exact test. The magnitude of the association was verified by the prevalence ratio, to a confidence level of 95%, for each of the independent categorical nominal variables concerning a dependent variable. Multivariate analysis was developed to identify the factors associated with UI, employing Poisson’s regression with robust variance to a confidence level of 95%. Different theoretical models were analyzed, including all the variables that presented *p*>0.05 in the bivariate analysis. These variables were tested and included or excluded according to theoretical plausibility and collinearity. In addition, some variables were tested in the multivariate analysis as adjustment variables (i.e., age, number of births or number of pregnancies, mode of delivery) due to their theoretical significance.

## 3. Results

The study included 106 older women who practiced physical activity, with ages varying from 60 to 88 years, averaging 69.1 ± 5.69. The frequency of physical exercise was 3.76 ± 1.49 times a week. The main characteristics of the sample were: predominantly white race, eutrophic or overweight, married or living with a partner, who finished high school, with an income between 1 and 2 minimum wages (class E), with access to health services through private health plans, and who mostly did not smoke tobacco or drink alcoholic beverages. Regarding the clinical data of the sample, most did not present intestinal constipation and had experienced on average 3 ± 2.16 pregnancies and 2 ± 2.03 births. Concerning comorbidities, arterial hypertension was highlighted, along with nocturia, and 54.7% presented UI (95% CI: 63.3–461), of which stress incontinence (SI) was most common.

Among the older women interviewed, the ICIQ score varied between 0 and 17 points, with an average of 3.87 ± 4.87. The majority did not experience any negative impacts on QL. Table 1 shows the description of these data.

Regarding the bivariate analysis of the socioeconomic variables and clinical records, for physically active older women with UI, data presented a statistically significant association with education levels, medications, dizziness and loss of balance during ADL, and waking up during the night (Table 2).

In multivariate analysis, after testing all the variables differently, the proposed model demonstrated that the prevalence of UI was higher among women with dizziness/loss of balance (PR 1.48; CI 1.06–2.07) and who wake up during the night (PR 1.63; CI 1.05–2.55). This result was established after adjusting the model for education level and the number of births (Table 3).

## 4. Discussion

This study evaluated the factors associated with UI in older women who practiced regular PA. Regarding UI, the participants presented a high prevalence for this dysfunction, mainly stress incontinence. Dizziness/loss of balance and waking up at night (sleep disruption) were significantly associated with a higher prevalence of UI in the sample, and was independent of education levels and the number of births. Strategies directed to the prevention and care throughout the life of these active seniors could have a substantial effect on these findings. Although PA is a protection factor for several health outcomes, including UI [18], other biological, psychological, social, and cultural aspects can lead to this issue in the female older population [19]. It must be noted that aspects related to the aging process can lead to the reduction of periurethral fibers and consequently to a decrease in urethral closure pressure, which leads to urine losses, especially during increases in intra-abdominal pressure [13]. Previous studies in developed countries found a UI prevalence of 27.5–28.0% among physically active female older people [20,21]. These results were lower than those obtained herein, which is possibly due to other factors that diverged between the study samples, such as the higher education levels of participants in developed countries. Researchers have highlighted the impact of social determinants (such as education levels) on people’s health, as better health-related indicators are frequently associated with higher levels of education [22,23]. Nevertheless, more studies are required on the direct evaluation of social determinants of health and the relationship between physical activity levels and UI.

It was verified that waking up during the night is related to UI in physically active older women. The main reason reported for sleep disruption, was having to urinate during the night, which refers to nocturia (one of the urinary symptoms of urge or mixed UI, in the study participants) [24].

Factors associated with nocturia include a decrease in bladder capacity, an increase of urine production during the night, sleep disorders, and even the time spent in bed [25]. Although PA presents a critical positive impact on the quality of sleep of older people [26], sleep quality can be negatively affected in seniors who suffer from UI, despite the benefits of the regular practice of physical activities [27,28].

Faria, Menezes, Rodrigues, Ferreira, & Bolsas (2014), reported a relationship between nocturia and chronic issues such as diabetes, sleep apnea, kidney disease, and hypertension. Some of these factors are present in our sample and could have contributed to the results [29]. The presence of nocturia and urine losses can be triggered by medications commonly indicated for these comorbidities, such as diuretics [29]. These medications increase urine production, urinary frequency, and cause an overload of bladder capacity, which generate urge-related urine losses [30]. Another factor that relates UI and the use of medications involves using anti-hypertension drugs such as alpha-blockers, which relax the smooth urethral muscle, contributing to UI [31].

The use of medications has been described in scientific literature as a risk factor for UI, however, with the model utilized, the present study did not establish a significant relationship. Nevertheless, the use of medications and their combined use in the presence of multimorbidity can contribute to different types of UI in older women. This corroborates the higher presence of urge- and mixed-UI in the studied population, and multimorbidity. Older adults are usually affected by multiple chronic diseases and generally use medications [32]. Practicing regular PA can help control these issues, mitigating signs, symptoms, and functional deterioration of these pathologies [32], and reducing the use or dosage of medication [33]. The presence of chronic diseases is multifactorial and must be assessed carefully in older age groups.

Besides sleep disruption, dizziness and loss of balance during ADL were also associated with the presence of UI in physically active older women. Physiological mechanisms and aspects of the environmental context can influence this association. One of the hypotheses for this association is that, biomechanically, women with UI use the hip adductor muscles to substitute for the pelvic floor muscles as a strategy to prevent urine losses. This overburden on the synergist muscles, considered as secondary motor muscles, can lead to a failure of the hip in maintaining balance. There is less lateral support, causing imbalance due to a minimized strategy of the hip to recover balance [34]. Moraes et al. (2018) remark that the older population is more vulnerable. As years progress, there can be a loss of strength and resistance, reducing physiological function, which decreases sensorial perception and leads to loss of balance [35,36].

Rosa and Braz (2016) carried out an integrative review that analyzed the risk of falls in older people with UI, and found an increased risk in this age group, especially when getting up to use the bathroom. External factors, such as lighting, the type of floor, and obstacles throughout the house, can be involved in the process of getting up from bed to use the bathroom and loss of balance during ADL [37].

The hypotheses considered herein on the relationships between UI, falls and loss of balance, can have significant clinical implications in the selection of strategies for the professional care of older people. These findings can enable more specific actions to be directed to the promotion, prevention, diagnosis, care, and rehabilitation of older adults that experience loss of balance and UI, which can prevent falls and their negative consequences.

On the number of births, scientific literature has extensive documentation that birth can increase the risk of UI. During birth, the pelvic floor muscles are subject to a fetal cephalic diameter four times larger than the urogenital diameter, which entails excessive stretching of the pelvic floor muscles during expulsion, leading to traumas and/or spontaneous perineal lacerations as well as episiotomy [38]. Physiological and anatomic adaptations during birth can cause changes to the functions of the pelvic floor muscles, decreasing muscle strength as well as control, coordination, tonus, resistance, and causing reflex-dysfunctions in the pelvic floor, mainly UI [39,40].

The coordination of voluntary movement is one of the functions of the musculoskeletal system, which activates the correct muscles at the right time and with adequate intensity to develop a specific action, and is considered adequate when the pelvic floor contracts without involving other muscles. Impaired function occurs when synergist muscles are activated (e.g., adductors, glutes, abdominals, respiratory) or the Valsalva maneuver is employed to maintain continence [40,41], which could be another factor for the presence of loss of balance in the older population. However, differently from the literature, this study demonstrated that the number of births did not influence the determination of UI, which can be explained by the low variability of the sample.

The average age of the group was over 69 years old, and the sample of older women over 80 years old consisted of only 6 participants. In this way, the results obtained herein must be interpreted with caution regarding the older age groups. Most participants were not obese (eutrophic and overweight) [42,43]. This reinforces the hypothesis of the importance of PA for weight control, even though the present study cannot provide a cause–effect relationship.

The analyses of social determinants, such as race, marital status, education level, and income, presented relevant samples in all stratification groups. The prevalence ratio assessment verified that low education levels were a risk factor for UI, although not statistically significant in the model utilized. Low levels of education can affect essential aspects, such as personal hygiene and other self-care activities of women [44]. Besides, low education levels can be related (directly or indirectly) to other important risk factors for UI: number of pregnancies, obesity, and healthy habits (diet and PA) [45]. Inter-sector public policies can promote better education and potentialize the empowerment and self-care of older women to address questions related to UI.

Some of these social determinants can be modified through public policies and better distribution of income and access to health services, and can interfere positively with older people’s health [45]. Most older women did not habitually consume alcohol and did not smoke. Health habits can positively affect the overall health condition and perception of the quality of life of older adults. In the case of non-modifiable social determinants (e.g., race), the majority were white older women, which corroborates the findings of Marques et al. [45]. Positive strategies related to social support between older people are valuable resources for maintaining a good quality of life. These strategies can positively influence the management and perception of adverse outcomes in the health of seniors [46]. Broader and more complex approaches must be considered by health professionals, involving social determinants in the care of older adults.

On the limitations of this study, due to its cross-sectional methodological design, no conclusions can be drawn related to causality, only to association; which reflects the clinical implications on the management and elaboration of assistance care programs, and construction of public policies for this age group. Despite the regular physical activity of older women, the PA intensity and duration were not measured; this hinders the determination of a more precise relationship between physical exercise and UI, or with the other evaluated parameters. Convenience sampling was used (non-probabilistic) and therefore, the application of the results obtained herein must be made with caution due to the lack of control of random error.

## 5. Conclusions

The study assessed the factors associated with urinary incontinence in active older women who participate in community groups. In physically active older women with UI, there was a significant association between waking up during the night and dizziness and loss of balance during ADL, regardless of education levels and the number of births. These findings can help build public policies for health promotion, the prevention of modifiable social determinants, and improvement of clinical programs directed to address UI in the female older population and modifiable social determinants.

## Figures and Tables

**Table 1 ijerph-18-05878-t001:** Description of participants. Natal/Brazil, 2020.

	*N*	%
Age		
60–69 years old	60	56.6
70–79 years old	40	37.7
80 years old and over	6	5.7
Body Mass Index		
Underweight (up to 18.5)	2	1.9
Eutrophic (18.5–24.9)	34	32.1
Overweight (25.0–29.9)	40	37.7
Obese (30.0 or more)	30	28.3
Ethnicity/Race		
While	56	52.8
*Parda*	40	37.7
Black	10	5.9
Marital status		
Single	17	16.0
Married/Living with partner	42	39.6
Divorced	16	15.1
Widow	31	29.2
Education level		
Illiterate	4	3.8
Incomplete fundamental level	19	17.9
Complete fundamental level	18	17.0
Incomplete secondary level	13	12.3
Complete secondary level	34	32.1
Complete graduate level	18	17.0
Income		
1–2 minimum wages	58	54.7
3–4 minimum wages	27	25.5
>4 minimum wages	14	13.2
Not informed	7	6.6
Access to health services		
Public	48	45.3
Private	56	52.8
No answer	2	1.9
Consumption of tobacco		
Yes	8	7.5
No	98	92.5
Consumption of alcohol		
Yes	21	19.8
No	85	80.2
Constipation		
Yes	39	36.8
No	67	63.2
Number of pregnancies		
Up to 1	19	17.9
2 or more	87	82.1
Number of births		
Up to 1	22	20.8
2 or more	84	79.2
Mode of delivery		
Vaginal delivery	49	46.2
Cesarean delivery	17	16.0
Both (vaginal and cesarean delivery)	28	26.4
Comorbidities		
Diabetes	19	17.9
Hypertension	37	34.9
Arthritis/Arthrosis	36	34.0
Others	1	0.9
None	13	12.3
Multimorbidity		
2 or more chronic diseases	53	50.0
1 chronic disease	40	37.7
None	13	12.3
Medication		
Yes	90	84.9
Noo	16	15.1
Restful sleep		
Little	31	29.2
Moderate	33	31.1
Complete	42	39.6
Dizziness and loss of balance during ADL		
Yes	42	39.6
No	64	60.4
Falls due to dizziness or loss of balance during ADL		
Yes	16	15.1
No	90	84.9
Urinary Incontinence		
Yes	58	54.7
No	48	45.3
Type of Urinary Incontinence		
Stress UI	43	40.6
Urge UI	11	10.4
Mixed UI	4	3.8
No IU	48	45.3
Waking up frequently during the night		
Yes	67	63.2
No	39	36.8
Reason for waking up during the night		
None	32	30.2
Change sleep position	3	2.8
Use bathroom	58	54.7
Noise	2	1.9
Other	10	9.4
No answer	1	0.9
ICIQ-SF		
Slight impact (1 to 5 points)	23	21.7
Moderate impact (6 to 12 points)	29	27.4
Severe impact (13 to 18 points)	6	5.7
Very severe impact (19 to 21 points)	0	0.0

ADL: activities of daily living; ICIQ-SF: International Consultation Incontinence Questionnaire Short Form.

**Table 2 ijerph-18-05878-t002:** Bivariate analysis between socioeconomic characteristics, clinical records, and urinary incontinence in older women who practice physical exercise. Natal, Brazil 2020.

Urinary Incontinence							
	Present		Absent		*p*	PR	CI (95%)
	*n*	%	*n*	%			
Age							
80 or more	4	66.6	2	33.3	0.479	1.14	0.55–1.20
70–79 years old	19	47.5	21	52.5		0.81	0.62–2.09
60–69 years old	35	58.3	25	41.6		1	
Ethnicity/Race							
Other	32	64.0	18	36.0	0.073	1.37	0.96–1.95
White	26	46.4	30	53.6		1	
Marital status							
Other	35	54.7	29	45.3	0.994	0.99	0.69–1.42
Married	23	54.8	19	45.2		1	
Education level							
Iliterate/primary	28	68.3	13	31.7	0.023 *	1.47	1.05–2.07
Secondary/Graduate studies	30	46.2	35	53.8		1	
Access to health services							
Public	31	62.0	19	38.0	0.158	1.28	0.90–1.82
Private	27	48.2	29	51.8		1	
BMI							
Overweight/Obese	37	52.9	33	47.1	0.587	0.90	0.63–1.29
Underweight/eutrophic	21	58.3	15	41.7		1	
Number of pregnancies							
2 or more	51	58.6	36	41.4	0.237	2.05	0.62–6.75
Up to 1	2	28.6	5	71.4		1	
Number of births							
2 or more	49	58.3	35	41.7	0.346	1.45	0.66–3.19
Up to 1	4	40.0	6	60.0		1	
Mode of delivery							
vaginal delivery	28	57.1	21	42.9	0.142	0.97	0.60–1.55
both (vaginal and cesarean delivery)	15	53.6	13	46.4		0.91	0.52–1.54
cesarean delivery	10	58.8	7	41.2		1	0.39–0.87
Multimorbidity							
1 or more chronic diseases	53	57.0	40	43.0	0.280	1.48	0.72–3.02
No chronic diseases	5	38.5	8	61.5		1	
Medications							
Yes	46	51.1	44	48.9	0.031 *	0.68	0.48–0.96
No	12	75.0	4	25.0		1	
Constipation							
Yes	22	56.4	17	43.6	0.789	1.04	0.73–1.49
No	36	53.7	31	46.3		1	
Consumption of tobacco							
Yes/ex-smoker	5	33.3	10	66.7	0.139	0.57	0.27–1.19
No	53	58.2	38	41.8		1	
Consumption of alcohol							
Yes	11	53.4	10	47.6	0.815	0.94	0.60–1.48
No	47	55.3	38	44.7		1	
Dizziness and loss of balance during ADL							
Yes	30	71.4	12	28.6	0.005 *	1.63	1.16–2.29
No	28	43.7	36	56.3		1	
Falls due to dizziness or loss of balance during ADL							
Yes	11	68.7	5	31.3	0.164	1.31	0.89–1.93
No	47	52.2	43	47.8		1	
Wakes up frequently during the night							
Yes	44	65.7	23	34.3	0.009 *	1.82	1.15–2.88
No	14	35.9	25	64.1		1	

* significant for *p* < 0.05; BMI: Body Mass Index.

**Table 3 ijerph-18-05878-t003:** Non-adjusted and adjusted prevalence and prevalence ratios for UI in older women who practice physical activities. Natal, Brazil, 2020.

	Urinary Incontinence									
	Absent		Present		Non-Adjusted		Adjusted			
	*N*	%	*n*	%	*p*	PR	CI (95%)	*p*	PR	CI (95%)
Dizziness and loss of balance during ADL										
Yes	12	28.6	30	71.4	0.005	1.63	1.16–2.29	0.006	1.63	1.15–2.30
Wake up frequently during the night										
Yes	23	34.3	44	65.7	0.009	1.82	1.15–2.88	0.030	1.70	1.05–2.76
Education level										
Illiterate/primary	13	31.7	28	68.3	0.023	1.47	1.05–2.07	0.483	1.12	0.80–1.57
Nº of births										
2 or more	35	41.6	53	56.4	0.346	1.30	0.66–3.19	0.573	1.37	0.45–4.21

PR: Prevalence ratio; CI: Confidence interval.

## Data Availability

The data presented in this study are available on request from the corresponding author.

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
