# Peer review of "Urinary Incontinence in Physically Active Older Women of Northeast Brazil"

_ijerph, 2021, doi:10.3390/ijerph18115878_

Round 1
Reviewer 1 Report
Thank you for asking me to review this paper, which uses a small sample of women in Brazil. Although a worthy subject for investigation, the paper will require significant improvement before publication.
Firstly, the language throughout is unclear and needs careful editing by a fluent English speaker.
From a scientific perspective there are several areas to address.
I cannot see a sample size calculation or any justification for the sample size chosen. In addition, the sample is selected from a group of physically active women, with no control group. Can the authors comment on this? It seems that this is merely a descriptive study of a small sample who happen to share a characteristic. Other epidemiological trials, such as EPIC, have included almost 20,000 participants.
Parity and gravidity are mentioned, but not mode of delivery, obstetric interventions, or obstetric injury. Clearly these are relevant factors and should be included.
How was "dizziness and loss of balance during ADL" selected as a variate? Which (and how many) other symptoms were assessed? Have they all been reported in the manuscript?
Fundamentally I am unclear what the aim of the paper was - as an epidemiological study the numbers are too small to draw meaningful conclusions, and as an observational study looking at the effect of physical activity, it lacks a control group.
Author Response
Comment 1: Firstly, the language throughout is unclear and needs careful editing by a fluent English speaker.
The manuscript has been thoroughly checked by a native professional English proofreader.
Comment 2: I cannot see a sample size calculation or any justification for the sample size chosen. In addition, the sample is selected from a group of physically active women, with no control group. Can the authors comment on this? It seems that this is merely a descriptive study of a small sample who happen to share a characteristic. Other epidemiological trials, such as EPIC, have included almost 20,000 participants.
The sample size was calculated from a sample for difference of proportions, and therefore the sample of 106 individuals can identify a 17% difference in proportion, with a confidence level of 95% and 80% test power (Newcombe, 1998). This information was added in the Methodology.
Because this is an association (observational) study, there is no control group.
Comment 3: Parity and gravidity are mentioned, but not mode of delivery, obstetric interventions, or obstetric injury. Clearly these are relevant factors and should be included.
The variable “delivery mode” had been analyzed in the paper, however because the pregnancy itself was already a risk factor for UI, independent of the delivery mode (Sangsawang, 2013), we chose not to include it initially. In addition, its inclusion did not interfere significantly with any result, nor with the multivariate model fit – however, we have included delivery mode in the revised version of the paper.
Regarding obstetric interventions and obstetric injuries, both are pertinent. Nevertheless, it is challenging to include, in a epidemiological study, all the possible variables that could interfere with the studied outcome.
Comment 4: How was "dizziness and loss of balance during ADL" selected as a variate? Which (and how many) other symptoms were assessed? Have they all been reported in the manuscript?
The variable dizziness and loss of balance during ADL is part of the section Postural Balance within the semi-structured questionnaire formulated with the researchers to verify associations between pelvic dysfunctions, such as UI, and postural balance.
The variables were: dizziness and loss of balance during ADL (have you felt dizzy or lost your balance in daily activities?) and falls due to dizziness or loss of balance during ADL (have you fallen due to dizziness or loss of balance during daily activities?).
The first variable was adequately described in the paper. The second was described in the methodology section, but it was mistakenly written in the results as “Risk of falling due to dizziness and loss of balance during ADL”. All corrections have been implemented throughout the new, revised version of the paper.

Reviewer 2 Report
- Reference of the first 3 sentences of the introduction does not seem to be correct, since Souza et al., em 2018, cite other references (United Nations. World population prospects: key findings and advance tables. The 2017 re- vision. New York: United Nations; 2017 & United Nations. World population prospects: key findings and advance tables. The 2015 re- vision. New York: United Nations; 2015).
- The reference about Active Aging is not adequate. I would cite the WHO (Active Ageing: A Policy Framework, 2002).
- The reference about UI prevalence is outdated. According to the International Continence Society, in 2017, the UI prevalence varies between 25 to 45% in women.
- Justification of the study is controversial. First the authors said that the studies about the theme is limited and this statement is followed by the information that have studies, but in scenarios different from Northeast.
- I would like to understand a little bit more about these community groups. The authors know how many community groups have in Natal? How many women attend each of these two groups? How did you choose these two community groups?
- Was there a sample size calculation? How you define the sample size?
- Exclusion criteria include women that with mobility difficulties, but those women were able to practice physical active. It sounds a little bit confusing for me. Is it possible to better explain this exclusion criteria?
- Number of pregnancies and births are obstetric history.
- Did you investigate just diabetes and hypertension as comorbidities? I saw in the results that you investigated other. It should be clear in the methods.
- What questions were answered using the perception of the interviewee? How was that?
- The references about the ICIQ-SF seems not adequate. For example, WHO 2002 reference do not say about severity of UI.
- Were all variables assessed in the multivariate model?
- One important issue about the methods is that the authors investigated UI in older women who practice physical activity, but they did not use any validated questionnaire to asses the level of physical activity, like the IPAQ. In my opinion, this fact weakens the study.
- The semi-structured questionnaire had a question about UI and they used the ICIQ-SF as measurement of UI. I would chose just one instrument to assess the presence of UI. For me, it does not make any sense to report the average score of the ICIQ-SF including those women that did not have UI. Considering that approximately 45% of the sample did not report urinary loss, the average was low (3.87) and did not represent the average score of those who reported urinary loss.
- What data about UI was used to investigate the associations in the bivariate and univariate analyses? From semi-structured questionnaire or from ICIQ-SF. How the authors assessed the outcome (UI) is not clear.
- Table 2 shows that medication is also associated with UI, but was not describe in the text.
- I did not understand what were the criterions used to chose the variables to include in the multivariate model. Why use number os births and do not use medication that was associated in the bivariate analysis? Did the authors use a conceptual framework for this?
- The objective of the study in the introduction is a little bit different of the objective described in the first phrase of the discussion.
- In the line 193, the correct form is “These results were LOWER…”. And is not clear how the educational level can explains this difference.
- Is not adequate to report results from bivariate analysis in the conclusion.
Author Response
Comment 1: Reference of the first 3 sentences of the introduction does not seem to be correct, since Souza et al., em 2018, cite other references (United Nations. World population prospects: key findings and advance tables. The 2017 re- vision. New York: United Nations; 2017 & United Nations. World population prospects: key findings and advance tables. The 2015 re- vision. New York: United Nations; 2015).
The authors respectfully disagree. The citation contemplates the epidemiological data in the contexts of global population and United Nations as well as the Brazilian population. This can be seen in the excerpt:
“..In Brazil, 13% of the population is over 60 years of age2, with the proportion expected to reach 29.3% by 20503.” (Souza et al. 2018).
As well as in the citations mentioned by the paper: ref. 2: “The 2017 re- vision. New York: United Nations; 2017 & United Nations” – data on the Brazilian population is given in page 17.
Ref. 3: “World population prospects: key findings and advance tables. The 2015 re- vision. New York: United Nations; 2015” – data on the Brazilian population is given in page 27.
Comment 2: The reference about Active Aging is not adequate. I would cite the WHO (Active Ageing: A Policy Framework, 2002).
The authors thank the reviewer for the suggestion and the reference has been included for this topic.
Comment 3: The reference about UI prevalence is outdated. According to the International Continence Society, in 2017, the UI prevalence varies between 25 to 45% in women.
The authors have included a current reference on this topic.
Comment 4: Justification of the study is controversial. First the authors said that the studies about the theme is limited and this statement is followed by the information that have studies, but in scenarios different from Northeast.
The authors agree and the sentence has been revised and modified to:” The majority of studies on factors associated with UI in physically active older people have been carried out in developed countries, with different socioeconomic and health-related characteristics from the Brazilian Northeast. Therefore, the objective of this study is to identify the factors associated with UI in physically active older women living in Northeast Brazil.”
Comment 5: I would like to understand a little bit more about these community groups. The authors know how many community groups have in Natal? How many women attend each of these two groups? How did you choose these two community groups?
The community groups selected were: Associação Rio Grandense Pro Idosos and the Centro de Promoção a saúde do Idoso.
Both present common characteristics such as contribute to the social wellbeing of older people by promoting physical exercise, collaborate to the development of social values, besides promoting participation in educative activities, to name a few. In addition, the older people do not reside at these centers.
Other institutions of Natal are more like nursing homes (residences) and therefore were not selected, as the focus of our study was to investigate a specific profile of physically active older women.
Most participants of both groups are women, and the amount of participants varies because the older people do not live there.
Comment Was there a sample size calculation? How you define the sample size?
As mentioned previously, the sample size was calculated from a sample for difference of proportions, and therefore the sample of 106 individuals is capable of identifying a 17% difference in proportion, with a confidence level of 95% and 80% test power (Newcombe, 1998). This information was added in the manuscript.
Comment Exclusion criteria include women that with mobility difficulties, but those women were able to practice physical active. It sounds a little bit confusing for me. Is it possible to better explain this exclusion criteria?
The authors agree and the sentence has been revised and modified to: “The exclusion criteria were: women with cognitive deficit to prevent the understanding of the questionnaires applied during the interviews; women who could not attend the interview, women who, for any reason, did not complete the questionnaires, and those who did not want to participate in the study.”
Comment 8: Number of pregnancies and births are obstetric history.
The authors agree and the expression has been modified.
Comment 9: Did you investigate just diabetes and hypertension as comorbidities? I saw in the results that you investigated other. It should be clear in the methods.
The authors agree and the sentence has been revised and modified to: ”presence of comorbidities or multimorbidities (diabetes, systemic arterial pressure, arthritis/arthrosis)”.
Comment 10: What questions were answered using the perception of the interviewee? How was that?
The authors understand that the term utilized was not sufficiently clear. We have corrected the text to refer to self-reporting by the participants only.
Comment 11: The references about the ICIQ-SF seems not adequate. For example, WHO 2002 reference do not say about severity of UI.
The authors agree. A typo has been identified, and the correct citation has been inserted in this topic: Klovning, A.;Avery, K.; Sandvik, H., Hunskaar, S. Comparison of Two Questionnaires for Assessing the Severity of Urinary Incontinence: The ICIQ-UI SF Versus the Incontinence Severity Index. Neurourology and Urodynamics. 2009, 28:411–415.
Comment 12: Were all variables assessed in the multivariate model?
Different theoretical models were analyzed, including all the variables that presented p>0.05 in bivariate analysis (education level, use of medication, dizziness and loss of balance during ADL and waking up frequently during the night). However only those that presented better fit by Omnibus test and deviance remained in the tested model. Other relevant variables established by literature were also included (number of births), which contributed to the fit of the final model.
Comment 13: One important issue about the methods is that the authors investigated UI in older women who practice physical activity, but they did not use any validated questionnaire to assess the level of physical activity, like the IPAQ. In my opinion, this fact weakens the study.
In this sense, the criterion employed to evaluate the practice of physical activity of participants is based on the questions within Module P – Lifestyles, from the questionnaire of the National Health Research, which addresses research involving participants from the Brazilian population (http://svs.aids.gov.br/dantps/acesso-a-informacao/inqueritos-de-saude/pns/2013/questionario/modulo-P.pdf). The following reference is cited in the manuscript: Szwarcwald, C.L.; Malta, D.C.; Pereira, C.A. et al National Health Survey in Brazil: design and methodology of application. Ciência & Saúde Coletiva. 2014, 19(2):333-342.
Comment 14: The semi-structured questionnaire had a question about UI and they used the ICIQ-SF as measurement of UI. I would chose just one instrument to assess the presence of UI. For me, it does not make any sense to report the average score of the ICIQ-SF including those women that did not have UI. Considering that approximately 45% of the sample did not report urinary loss, the average was low (3.87) and did not represent the average score of those who reported urinary loss.
Measurement of UI as outcome measure was carried out by the ICIQ-SF questionnaire, as described in the methodology. The question of the semi-structured questionnaire has the objective of characterizing the sample, especially regarding the type of UI. We have improved the clarity of the sentence, and reformulated to only “type of UI”.
Regarding the score of ICIQ-SF for women with no UI reported in the study, the authors agree and the information was removed.
Comment 15: What data about UI was used to investigate the associations in the bivariate and univariate analyses? From semi-structured questionnaire or from ICIQ-SF. How the authors assessed the outcome (UI) is not clear.
UI data utilized to investigate the associations were: presence of absence of UI. For both the univariate and bivariate analyses, this information was verified from the ICIQ-SF questionnaire, utilized for the specific measurement of this outcome as mentioned in the methodology.
As previously explained, the question of the semi-structured questionnaire has the objective of characterizing the sample, especially regarding the type of UI. We have improved the clarity of the sentence, and reformulated to only “type of UI”.
Comment 16: Table 2 shows that medication is also associated with UI, but was not describe in the text.
The authors agree and the description has been added to the text.
Comment 17: I did not understand what were the criterions used to chose the variables to include in the multivariate model. Why use number os births and do not use medication that was associated in the bivariate analysis? Did the authors use a conceptual framework for this?
As mentioned previously, the authors tested the theoretical model from the statistically significant variables of the bivariate analysis. There was theoretical plausibility for the variable use of medication in this model and, therefore, we also tested this variable, but it did not adjust the multivariate model. Thus, after verifying the theoretical models proposed, we opted to remove the variable use of medication and insert the number of births, respecting the fit of the theoretical model proposed for the studied sample. Finally, we found 3 multivariate models but we selected the best fitted model.
Comment 18: The objective of the study in the introduction is a little bit different of the objective described in the first phrase of the discussion.
The authors agree and the sentence has been revised and modified to: “This study evaluated the associated factors with UI in older women who practiced regular PA.”
Comment 19: In the line 193, the correct form is “These results were LOWER…”. And is not clear how the educational level can explains this difference.
The sentence has been revised and modified to: “these results were lower than those obtained herein, which is possibly due to other factors that diverged between the study samples, such as the higher education levels of participants in developed countries. Researchers have highlighted the impact of social determinants (such as education levels) on people's health, as better health-related indicators are frequently associated with higher levels of education [18,19]. Nevertheless, more studies are required on the direct evaluation of social determinants of health and the relationship between physical activity levels and UI.”.
Comment 20: Is not adequate to report results from bivariate analysis in the conclusion.
The authors agree and the sentence has been revised and modified to: ”The study assessed the factors associated with urinary incontinence in active older women who participate in community groups. In physically active older women with UI, there was a significant association between waking up during the night and dizziness and loss of balance during ADL, regardless of education levels and number of births. These findings can help build public policies for health promotion, prevention of modifiable social determinants, and improvement of clinical programs directed to address UI in the female older population and modifiable social determinants”.
The authors wish to thank you for the time dedicated to reviewing our manuscript. We believe the new, revised version, is much better. The manuscript has undergone a throughout, detailed review to improve clarity and understandability. Please check the changes implemented, we are now happy with this reviewed, revised version of the manuscript.

Round 2
Reviewer 1 Report
Thank you for the opportunity to review the revision of this manuscript. I thank the authors for their hard work in responding to the first review.
Overall the paper is stronger than it was. All of the concerns raised have been appropriately addressed within the manuscript.
Author Response
Dear Reviewer,
Thank you very much for your revision. Your feedback helped us to improve the manuscript. Best regards.
Reviewer 2 Report
- I strongly disagree of the authors’ response to my first comment about the reference of the three first phrases of the introduction. As the authors said, Sousa and colleagues cited other references (As well as in the citations mentioned by the paper: ref. 2: “The 2017 re- vision. New York: United Nations; 2017 & United Nations” – data on the Brazilian population is given in page 17.
Ref. 3: “World population prospects: key findings and advance tables. The 2015 re- vision. New York: United Nations; 2015” – data on the Brazilian population is given in page 27) and those are the references that the authors would have to cite.
- About my comment regarding the community groups, I think it is important to include why the authors selected those two community groups, from the universe of community groups in Natal. That can enhance the information about sampling selection. Another point that deserves clarification is about the sentence “The sample was non-probabilistic, randomly selected…”. Non-probabilistic sampling involves non-random selection based on convenience or other criteria, allowing the researches to easily collect data. Thus, I can not understand how the authors have a non-probabilistic sample, randomly selected. The information about sampling selection needs to be very clear.
- About my 12th and 17th comments, according to the authors’ answers, they have a conceptual framework, for example, they included number of births as a adjustment variable. This information (variables that were used as adjustment) is important to put in the methods section. In addition, did not seems plausible for me to retain or remove variables of the final model based only in the Omnibus test. If you have a theoretical justification to include all those variable in the bivariate analysis, it does not make sense to remove a variable based on better fitted model only. I would like to see the output of the three models and the Omnibus test of them. I think it needs to be clear in the analysis plan all the steps to reach the final model. For me is not sufficient to say “In multivariate analysis, after testing all the variables differently, the proposed model demonstrated…” in the results section.
- Regarding my 13th comment, ok. The authors used questions from the National Health Survey as reference. But is not clear how they defined physically active women. A physically active women was defined considering just the first question about physical activity (P34 in the questionnaire)? This definition needs to be very clear in the methods section, since they are studying women physically active. Just sporadically participate of the community group, without a frequency and duration of the activity that define a physically active women, is not enough.
Author Response
Comment 1: - I strongly disagree of the authors’ response to my first comment about the reference of the three first phrases of the introduction. As the authors said, Sousa and colleagues cited other references (As well as in the citations mentioned by the paper: ref. 2: “The 2017 re- vision. New York: United Nations; 2017 & United Nations” – data on the Brazilian population is given in page 17. Ref. 3: “World population prospects: key findings and advance tables. The 2015 re- vision. New York: United Nations; 2015” – data on the Brazilian population is given in page 27) and those are the references that the authors would have to cite.
The authors thank the reviewer for the suggestion. These references have been included in the Introduction.
Comment 2: A-: About my comment regarding the community groups, I think it is important to include why the authors selected those two community groups, from the universe of community groups in Natal. That can enhance the information about sampling selection. B-: Another point that deserves clarification is about the sentence “The sample was non-probabilistic, randomly selected…”. Non-probabilistic sampling involves non-random selection based on convenience or other criteria, allowing the researchers to easily collect data. Thus, I cannot understand how the authors have a non-probabilistic sample, randomly selected. The information about sampling selection needs to be very clear.
The authors thank the reviewer for the suggestion. The following information has been added in the manuscript: “A non-probabilistic sample of older women who practiced regular physical exercise in two community groups of Natal, Northeast Brazil, was obtained. The senior groups were selected for convenience; both offered regular, low-impact supervised physical activity five times a week: water aerobics, dancing and gymnastics routines, for 60 minutes each”.
Comment 3: About my 12th and 17th comments, according to the authors’ answers, they have a conceptual framework, for example, they included number of births as a adjustment variable. This information (variables that were used as adjustment) is important to put in the methods section. In addition, did not seems plausible for me to retain or remove variables of the final model based only in the Omnibus test. If you have a theoretical justification to include all those variables in the bivariate analysis, it does not make sense to remove a variable based on better fitted model only. I would like to see the output of the three models and the Omnibus test of them. I think it needs to be clear in the analysis plan all the steps to reach the final model. For me is not sufficient to say “In multivariate analysis, after testing all the variables differently, the proposed model demonstrated…” in the results section.
The authors thank the reviewer for the suggestion and more information about the model selection process has been added in the manuscript, in the Material and Methods section:
<<Different theoretical models were analyzed, including all the variables that presented p>0.05 in the bivariate analysis. These variables were tested and included or excluded according to theoretical plausibility and collinearity. In addition, some variables were tested in the multivariate analysis as adjustment variables (i.e. age, number of births or number of pregnancies, mode of delivery) due to its theoretical significance>>.
Comment 4: Regarding my 13th comment, ok. The authors used questions from the National Health Survey as reference. But is not clear how they defined physically active women. A physically active women was defined considering just the first question about physical activity (P34 in the questionnaire)? This definition needs to be very clear in the methods section, since they are studying women physically active. Just sporadically participate of the community group, without a frequency and duration of the activity that define a physically active women, is not enough.
The authors agree and this information has been added in the manuscript: “According to this questionnaire, physically active women were defined as those who practiced any type of physical exercise or sport in the last three months (not considering physiotherapy).”. You can find below the original question, in portuguese:
P34. Nos últimos três meses, o(a) sr(a) praticou algum tipo de exercício físico ou esporte? (não considere fisioterapia) 1. Sim 2. Não